# Automated Efficient Estimation using Monte Carlo Efficient Influence Functions

**Raj Agrawal**
Basis Research Institute, Broad Institute
`raj@basis.ai`

**Sam Witty**
Basis Research Institute, Broad Institute
`sam@basis.ai`

**Andy Zane**
Basis Research Institute, UMass Amherst
`andy@basis.ai`

**Eli Bingham**
Basis Research Institute, Broad Institute
`eli@basis.ai`

## Abstract

Many practical problems involve estimating low dimensional statistical quantities with high-dimensional models and datasets. Several approaches address these estimation tasks based on the theory of influence functions, such as debiased/double ML or targeted minimum loss estimation. We introduce *Monte Carlo Efficient Influence Functions* (MC-EIF), a fully automated technique for approximating efficient influence functions that integrates seamlessly with existing differentiable probabilistic programming systems. MC-EIF automates efficient statistical estimation for a broad class of models and functionals that previously required rigorous custom analysis. We prove that MC-EIF is consistent, and that estimators using MC-EIF achieve optimal $\sqrt{N}$ convergence rates. We show empirically that estimators using MC-EIF are at parity with estimators using analytic EIFs. Finally, we present a novel capstone example using MC-EIF for optimal portfolio selection.

## 1 Introduction

Over the past several decades, there has been remarkable progress on robust and efficient statistical estimation, especially for high dimensional problems. A particularly compelling class of such methods are built on a foundation of *efficient influence functions* (EIF), i.e., functional derivatives in the space of probability distributions [Ken22]. These methods have been particularly fruitful in causal inference applications, where estimating quantities such as the average treatment effect require modeling high-dimensional nuisance parameters relating confounders to treatment and outcome variables. Intuitively, these methods focus finite statistical resources on quantities that matter, and not on nuisance parameters that only indirectly inform the statistical quantities we wish to estimate.

Despite their successes, estimation methods based on the EIF have lagged behind the kinds of automation that machine learning practitioners have grown accustomed to, instead requiring complex manual derivation on a case-by-case basis. This is contrasted with the generality of automatic differentiation (AD) systems [BPRS18] and probabilistic programming languages (PPLs) such as Pyro [BCJ$^+$19] or Gen [CTSLM19], which automate numerical computations for probabilistic inference. EIF-based estimators have historically eluded this level of automation and generality, because exact recovery of the EIF requires solving high-dimensional integral equations.

**Contributions.** We introduce *Monte Carlo Efficient Influence Functions* (MC-EIF), a general and automated technique for numerically computing EIFs using only quantities that are already available from existing AD and PPL systems. Our key insight is that EIFs can be expressed equivalently as a product of (i) the gradient of the functional, (ii) the inverse Fisher information matrix, and (iii) the

38th Conference on Neural Information Processing Systems (NeurIPS 2024).

gradient of the log-likelihood, as shown in Theorem 3.4 in Section 3. In Section 4, we show how MC-EIF can be used to automatically construct a variety of efficient estimators for a broad class of models and functionals, avoiding the need for complex manual and error-prone derivations.

In summary, we show that: (i) MC-EIF provides accurate estimates of the true EIF, enabling efficient estimation, and (ii) MC-EIF is very general, applying to many functionals and models that can be written as probabilistic programs. In Section 3, we introduce MC-EIF and provide a non-asymptotic error bound on the quality of our approximation. We show how estimators using MC-EIF achieve the same asymptotic guarantees as using analytic EIFs in Section 4. In Section 5, we show empirically that MC-EIF produces more accurate estimates of the EIF than existing automated approaches, and using MC-EIF as a drop-in replacement for the analytic EIF does not degrade estimation accuracy in a variety of benchmarks, including a novel capstone on optimal portfolio allocation.

**Related Work.** Influence function-based estimators have a rich history in the statistics and machine learning literature [BKB$^+$93, Tsi06, Ken22, HDDOV22]. Despite their effectiveness, these methods have historically required custom and complex mathematical analysis for specific combinations of models and functional. Even an incomplete recent survey of the influence function-based estimation literature yields a large collection of complex scenario-specific research. For targeted minimum loss estimation (TMLE) [VDLR06]; cluster-randomized trials [BvdLA$^+$23], continuous time-dependent interventions [RGvdL22], mixed experimental and observational data [DTA$^+$22], mediation analysis with longitudinal data [WvdLP$^+$23], subgroup treatment effect estimation [WPvdL$^+$23], survival and competing risks analysis [RvdL24], continuous time-to-event outcomes [REvdL23], and variable importance measures for effect estimates [LHvdL23]. For double/debiased machine learning [CCD$^+$18]; difference-in-differences [Cha20], instrumental variable designs [JTB21a], and mediation analysis [FHL$^+$22].. Importantly, our work does not introduce novel efficient estimators; instead it aims to lower the mathematical burden for practitioners who wish to use existing influence function-based efficient estimator templates (see Section 4) with custom models and/or functionals.

Our work is not the first to attempt to automate and generalize computations for efficient statistical estimation. Perhaps the closest technique we are aware of is approximating the influence function using finite differences on kernel-smoothed empirical distributions [FQWD15, CLvdL19, JWZ22a]. We provide a thorough comparison with this method in Section 5, demonstrating how MC-EIF automates and scales better to high dimensional problems, exactly the settings where efficient estimation is most useful. Recent work has made progress towards general efficient estimators, but still impose strong restrictions on models and/or functionals. DML-ID [JTB21b] extends double machine learning to nonparametric causal graphs and marginal density under intervention functionals. Similarly, the kernel debiased plugin estimator [CGMM23] implements a version of TMLE that bypasses the influence function computations for models defined in a RKHS. Finally, other recent work [CNS22, FS23, CNQMS21, CNQMS22] approximates the efficient influence function for generalized method-of-moment estimators.

Finally, we note that there are a number of intriguing connections between three related but distinct mathematical objects: the *efficient influence function* in semiparametric statistics [Tsi06], the *natural gradient* in information geometry [Ama16], and the so-called *empirical influence function* in robust statistics and machine learning [Law86]. A comprehensive review of these connections is beyond the scope of this paper, and we focus here on two particularly important points for contextualizing our work. First, we emphasize that **the efficient and empirical influence functions are not equivalent**: the efficient influence function is a fundamental mathematical object in semiparametric statistical theory which quantifies the effect of perturbing a functional in an arbitrary direction in the space of probability measures, while the empirical influence function is a distinct and somewhat more specialized [BNL$^+$22] object quantifying the effect of perturbing individual training points in a parametric statistical model. More specifically, in the parametric setting the efficient influence function is defined as shown in Theorem 3.4 in terms of the Fisher information matrix [Tsi06], whereas the empirical influence function is defined in terms of the Hessian of a model at the training points, two quantities with very different mathematical and statistical properties that are not straightforwardly interchangeable [KHB19]. Second, we note that **existing algorithms for computing natural gradients and empirical influence functions cannot immediately be adapted to efficient estimation**. Specifically, our algorithm described in Section 3 for Monte Carlo approximation of the efficient influence function is similar in structure to some previous algorithms developed for estimating the natural gradient [TR19] and empirical influence function [KL17, GSL$^+$19]. This would seem to suggest porting other methods that compute more heavily biased approximations of the natural gradi-

ent [GLB+18] and empirical influence function to computing the efficient influence function, as some of these methods are known to scale to even the largest neural network models deployed in practice today [GBA+23]. However, these approximations are not directly applicable in our setting because provably efficient estimation is only known to be possible with tight control over any approximation error introduced in computing the efficient influence function, as discussed in Section 4 below and in [JWZ22a]. Relaxing these restrictions to enable similarly scalable variations on the basic MC-EIF framework of Section 3 is an important direction for future work.

## 2   Problem Statement

**General Problem.** We consider the estimation of some estimand $\theta^* \in \mathbb{R}^L$, where $L \in \mathbb{N}$ denotes the dimension of the target quantity. Typically, we can express $\theta^* = \Psi(\mathbb{P}^*)$ for some known functional $\Psi$, where $\Psi$ maps a probability distribution to a vector in $\mathbb{R}^L$, and $\mathbb{P}^*(x)$ denotes the true data-generating distribution over some vector of observables $x \in \mathbb{R}^D$, $D \in \mathbb{N}$. Many estimation tasks involve high-dimensional *nuisance* parameters, or quantities of no immediate value to the analyst. For example, to estimate the average treatment effect, one might need to adjust for high-dimensional confounders.

**Semiparametric Solution.** Semiparametric statistics provides a mathematical framework for optimally estimating $\theta^*$ in the presence of potentially complex, high-dimensional nuisance parameters. A standard way to estimate $\theta^*$ is with the *plug-in approach*; construct an estimate $\hat{\mathbb{P}}$ of $\mathbb{P}^*$ and report $\hat{\theta} = \psi(\hat{\mathbb{P}})$. Unfortunately, the plug-in approach can lead to provably sub-optimal estimates of $\theta^*$ due to poor estimates of $\mathbb{P}^*$ [Tsi06, CCD+18, FS23]. Instead, a general recipe for efficiently estimating $\theta^*$ from finite data $\{x_n\}_{n=1}^N$, where $x_n \overset{\text{iid}}{\sim} \mathbb{P}^*(x)$ for $n = 1, \cdots, N$, is given by the following three-steps: (i) use $N/2$ samples to construct an initial estimate $\hat{\mathbb{P}}$ of $\mathbb{P}^*$, (ii) compute the influence function (to be defined shortly) of $\Psi$ at the estimate $\hat{\mathbb{P}}$, and (iii) evaluate the influence function at the held out $N/2$ datapoints to derive a corrected estimate.[1] In Section 4, we elaborate on how influence functions are used to construct several popular efficient estimators.

**Influence Functions.** A central premise of this paper is that to automate efficient estimation, it suffices to automate the computation of *influence functions*, which can be thought of as gradients in function space. We make this precise below.

**Definition 2.1.** (Gateaux derivative) Consider the $\epsilon$-perturbed probability distribution $\mathbb{P}_\epsilon := (1 - \epsilon)\mathbb{P} + \epsilon\mathbb{Q} = \mathbb{P} + \epsilon(\mathbb{Q} - \mathbb{P})$, where $\mathbb{Q}$ is some probability distribution. $\Psi$ is *Gateaux* differentiable at $\mathbb{Q}$ if the following limit exists:

$$\frac{d}{d\epsilon}\Psi(\mathbb{P}_\epsilon)\bigg|_{\epsilon=0} = \lim_{\epsilon \to 0} \frac{\Psi(\mathbb{P}_\epsilon) - \Psi(\mathbb{P})}{\epsilon}.$$

The Gateaux derivative can be viewed as a generalization of the directional derivative from ordinary calculus; it characterizes how much a functional changes at a point $\mathbb{P}$ in the direction $\mathbb{Q} - \mathbb{P}$.

**Definition 2.2.** (Influence function) Suppose there exists a square integrable function $\varphi \in L^2(\mathbb{P})$ such that

$$\frac{d}{d\epsilon}\Psi(\mathbb{P}_\epsilon)\bigg|_{\epsilon=0} = \langle \varphi, q - p \rangle_{L^2} = \int_{x \in \mathbb{R}^D} \varphi(x)(q(x) - p(x))dx$$

for all $\mathbb{Q} \in \mathcal{M}$ and $\mathbb{E}_{x \sim \mathbb{P}}[\varphi(x)] = 0$, where $\mathcal{M}$ denotes some space of probability distributions and $p(\cdot)$ and $q(\cdot)$ are the density functions for $\mathbb{Q}$ and $\mathbb{P}$, respectively. Then, $\varphi$ is called an influence function for $\Psi$ at $\mathbb{P}$.

An influence function is a re-centered "functional gradient" in $L^2(\mathbb{P})$: just as the Euclidean inner product between the gradient of a function and a vector yields the directional derivative in ordinary differential calculus, the $L^2(\mathbb{P})$ inner product between the influence function and perturbation $\mathbb{Q} - \mathbb{P}$ yields the Gateaux directional derivative. Influence functions, however, are not always unique [Tsi06, Ken16] — some may lead to higher asymptotic variance estimators than others. The optimal influence function minimizes asymptotic variance, and is called the *efficient influence function*

---

[1]There are some small variations to this three-step recipe such as using cross-fitting [CCD+18] instead of a simple equal split of the data; see also Section 4.

(EIF). When the EIF exists, it is $\mathbb{P}$ almost everywhere unique, and found through a Hilbert space projection onto what is known as the nuisance tangent space. We defer details to [Tsi06] and [Ken16].

As the influence function in Definition 2.2 is defined implicitly as a solution to an infinite set of integral constraints over $\mathcal{M}$, it is often hard to find. Entire papers have been written to analytically derive influence functions; see, for example, the papers listed in Section 1. For even experts in machine learning and statistics, such derivations are out-of-reach, time consuming, and error prone.

## 3 Monte Carlo Efficient Influence Function

Much of the work in semiparametric statistics and efficient estimation has focused on scenarios where the nuisance function is modeled nonparametrically [Tsi06, VDLR06, CCD$^+$18, Ken22]. However, practitioners often use high-dimensional parametric models such as generalized linear models, neural networks, and tensor splines in practice due to their flexibility and ability to scale to large datasets. Due to the richness of these high-dimensional spaces, inference is still statistically challenging and benefits from efficient estimation; see, for example, Table 1 in [CCD$^+$18]. Specifically, in contrast to traditional low-dimensional parametric models where maximum likelihood estimation is typically efficient [FR22, Rao45], high-dimensional parametric models often exhibit distinct asymptotic behaviors [vdV98, KBB$^+$13, HTW15]. In these high-dimensional models, estimates may converge slower than classic $O_p(\frac{1}{\sqrt{N}})$ rates without the application of efficient inference methods [VDLR06, CCD$^+$18, Ken22]. A key question we address is whether using a high-dimensional parametric model simplifies the process of solving Definition 2.2. We show that it does below.

**Notation.** We let $\phi \in \Phi \subset \mathbb{R}^p$ denote a finite-dimensional parameter specifying a distribution on the observed random variables $x \in \mathbb{R}^D$ for $p < \infty$, $p \in \mathbb{N}$. $\mathbb{P}_\phi(x)$ corresponds to a distribution in this space, and $\mathbb{P}_{\phi^*}(x)$ the true distribution, or the one closest to the true data-generating distribution in Kullback–Leibler distance. We let $\psi(\phi)$ denote a function $\mathbb{R}^p \mapsto \mathbb{R}^L$ that equals the evaluation of the functional $\Psi(\mathbb{P}_\phi)$ for all $\phi \in \Phi$. Under mild differentiability assumptions, we provide the analytic formula for the EIF in Theorem 3.4.

The first assumption states that the density of $\mathbb{P}_\phi(x)$ is continuous and differentiable with respect to $\phi$, a condition satisfied by many parametric model families. For example, the univariate Gaussian density $\frac{1}{\sqrt{2\pi}} \exp\left(-0.5(x - \phi)^2\right)$ is a continuous and differentiable function of its mean, $\phi \in \mathbb{R}$.

**Assumption 3.1.** $\forall x \in \mathbb{R}^D$, the map $\phi \mapsto \mathbb{P}_\phi(x)$ is continuous and differentiable with respect to $\phi$.

The next assumption is also satisfied for many functionals. For example, consider the mean functional $\Psi(\mathbb{P}_\phi) = \mathbb{E}_{x \sim \mathbb{P}_\phi}[x]$. Continuing with the univariate Gaussian example from above, where the mean is unknown, we have $\psi(\phi) = \phi$, which is a continuous and differentiable function of $\phi$.

**Assumption 3.2.** $\psi(\phi)$ is a continuous and differentiable function of $\phi$.

The last assumption requires that the Fisher information matrix be invertible, which is necessary for $\phi$ to be identifiable [Tsi06].

**Assumption 3.3.** Fisher information $I(\phi) \coloneqq \mathbb{E}_{x \sim \mathbb{P}_\phi(x)}[\nabla_\phi \log \mathbb{P}_\phi(x) \nabla_\phi \log \mathbb{P}_\phi(x)^T]$ is invertible.

**Theorem 3.4.** *(Theorem 3.5 in [Tsi06]) Suppose Assumption 3.1, Assumption 3.2, and Assumption 3.3 hold. Then, the efficient influence function $\varphi_\phi(\tilde{x})$ at $\phi$ evaluated at the point $\tilde{x} \in \mathbb{R}^D$ equals*

$$[\nabla_\phi \psi(\phi)]^T I(\phi)^{-1} \nabla_\phi \log \mathbb{P}_\phi(\tilde{x}). \tag{1}$$

While Equation 1 has been around for many decades, it has mainly been used as a theoretical tool for mathematical statisticians. In particular, Equation 1 is typically evaluated at the true data generating parameter $\phi^*$ to characterize the theoretical asymptotic variance of an estimator. In other instances, it is used to derive approximate confidence intervals; see, for example, Chapter 3 in [Tsi06]. In the following Sections, we discuss how Equation 1 provides a key ingredient in automating efficient estimation in high-dimensional parametric models.

### 3.1 Numerically Approximating the EIF

Given a model $\mathbb{P}_\phi(\cdot)$ and functional $\Psi(\cdot)$, we seek to automatically compute Equation 1. Our *Monte Carlo efficient influence function* (MC-EIF) estimator achieves this automation by replacing $\psi(\phi)$ and

$I(\phi)$, which are typically unknown, with stochastic approximations $\hat{\psi}_M(\phi)$ and $\hat{I}_M(\phi)$ computed from $M \in \mathbb{N}$ Monte Carlo samples:

$$\hat{\varphi}_{\phi,M}(\tilde{x}) := [\nabla_\phi \hat{\psi}_M(\phi)]^T \hat{I}_M(\phi)^{-1} \nabla_\phi \log \mathbb{P}_\phi(\tilde{x}). \tag{2}$$

Here, we show that Equation 2 leads to an automated and accurate approach to numerically computing EIFs using only quantities provided by existing AD and PPL systems.

**Approximating $\hat{I}_M(\phi)^{-1} \nabla_\phi \log \mathbb{P}_\phi(\tilde{x})$.** We draw $x_m \overset{\text{iid}}{\sim} \mathbb{P}_\phi(x), 1 \leq m \leq M$ for $M \in \mathbb{N}$, and let

$$\hat{I}_M(\phi) = \frac{1}{M} \sum_{m=1}^{M} \nabla_\phi \log \mathbb{P}_\phi(x_m) \nabla_\phi \log \mathbb{P}_\phi(x_m)^T. \tag{3}$$

A naive approach for computing $\hat{I}_M(\phi)^{-1} \nabla_\phi \log \mathbb{P}_\phi(\tilde{x})$ is calculating the full $p \times p$ matrix in Equation 3, inverting it, and then taking its product with the score vector $\nabla_\phi \log \mathbb{P}_\phi(x_m)^T \in \mathbb{R}^p$ computed from AD. This naive approach takes $O(Mp^2 + p^3)$ time and $O(p^2)$ memory which might be too expensive for large $p$. Instead, we exploit AD and numerical linear algebra techniques to avoid explicitly storing and inverting the approximate Fisher information matrix, similar to [KL17]. Suppose that we have a black-box method to compute Fisher vector products $\hat{I}_M(\phi)v$ for arbitrary vectors $v \in \mathbb{R}^p$. Then, we could use the conjugate gradient algorithm to iteratively find $\hat{I}_M(\phi)\nabla_\phi^{-1} \log \mathbb{P}_\phi(\tilde{x})$, where the cost of each conjugate gradient step is determined by the cost to compute $\hat{I}_M(\phi)v$. While the number of conjugate gradient steps needs to be $p$ for an exact inverse, often far fewer iterations are required for a close approximate solution [WPG$^+$19]. To make computing $\hat{I}_M(\phi)v$ efficient, we collect the $M$ simulated datapoints in the matrix $X_M \in \mathbb{R}^{M \times D}$ and let

$$\log \mathbb{P}_\phi(X_M) := (\log \mathbb{P}_\phi(x_1), \cdots, \log \mathbb{P}_\phi(x_M))^T \in \mathbb{R}^M.$$

Then, $\hat{I}_M(\phi)v$ equals

$$\left[ \frac{1}{M} J_M^T J_M \right] v = \left[ \frac{1}{M} J_M^T \right] [J_M v], \tag{4}$$

where $J_M = \nabla_\phi \log \mathbb{P}_\phi(X_M) \in \mathbb{R}^{M \times p}$ is the Jacobian matrix. We use *Pearlmutter's trick* to avoid computing the entire Jacobian matrix [Pea94]. In particular, this method allows us to compute the Jacobian vector product $v_M = [J_M v] \in \mathbb{R}^M$ in time proportional to a single evaluation of $\log \mathbb{P}_\phi(X)$ and $O(M + p)$ memory. Similarly, we use the vector Jacobian product to compute $J_M^T v_M$.

**Approximating $\nabla_\phi \hat{\psi}_M(\phi)$.** Robust estimation with MC-EIF does not require exact gradients. Instead, it only requires a sequence of gradient estimators $\{\nabla_\phi \hat{\psi}_m(\phi)\}_{m=1}^{\infty}$ of $\nabla_\phi \psi(\phi)$ whose error can be bounded above by some $\Delta_m > 0$, where the $M$th iterate $\nabla_\phi \hat{\psi}_M(\phi)$ is used in Equation 2.[2] Using such a sequence guarantees that the approximation error of Equation 2 is not dominated by $\nabla_\phi \hat{\psi}_M(\phi)$. In practice, the target functional $\psi(\theta)$ might be quite complex, making gradient estimation challenging. For example, it might involve taking expectations with respect to conditionals of $\mathbb{P}_\phi(x)$, or be defined implicitly as a solution to an optimization problem as in [JWZ22a].

One particularly simple and general way to address this challenge is to implement a Monte Carlo estimator of $\psi$ that can be transformed via automatic differentiation into an efficient Monte Carlo estimator for its gradient, a well-understood problem that is beyond the scope of this paper to review. We note that for the very wide class of functionals that can be written as nested expectations, recent work [RCY$^+$18, SW23, LHSM23] gives formal statements of smoothness assumptions and theoretical results sufficient to obtain the oracle rate $\Delta_m$ in terms of numbers of samples, as well as algorithms that can be implemented using automatic differentiation software like PyTorch [PGC$^+$17]. For example code snippets of functionals, see Appendix B.

### 3.2 Theoretical Guarantees for MC-EIF

We conclude by deriving a non-asymptotic error bound for how well Equation 2 approximates Equation 1. For fixed input dimension $D$ and model sizes $p$, Equation 2 converges to Equation 1 at a

---

[2]In Section 3.2, we require that $\Delta_m = o\left(\sqrt{m^{-1}p \log p}\right)$.

$O_p(1/\sqrt{M})$ rate by the Law of Large Numbers. As we are interested in high-dimensional parametric families, we analyze the behavior of our approximation as a function of both input dimension $D$ and model size $p$. To prove our result, we use standard tools and assumptions from empirical process theory such as the requirement of sub-Gaussian tails [vdV98].

**Assumption 3.5.** Suppose $x \sim \mathbb{P}_\phi(x)$. There exists a universal constant $0 < C_1 < \infty$ such that the normalized score vector $\tilde{x} := \frac{1}{\sqrt{D}} \nabla_\phi \log \mathbb{P}_\phi(x)$ is a sub-Gaussian random vector with parameter $C_1$.

As $\mathbb{E}[\|\nabla_{\phi_j} \log \mathbb{P}_\phi(x)\|_2^2] = O(D)$, $1 \leq j \leq p$, the division by $\sqrt{D}$ in Assumption 3.5 ensures that the variance of the score does not grow unboundedly as $D \to \infty$. Thus, our assumption that $\tilde{x}$ is sub-Gaussian is mild. Assumption 3.6 below ensures that the functional and score are smooth enough by bounding their gradients.

**Assumption 3.6.** There exist universal constants $C_2, C_3 < \infty$ such that $\|\nabla_\phi \psi(\phi)\|_F < C_2$ and $\left\| \frac{\nabla_\phi \log \mathbb{P}_\phi(x^*)}{D} \right\|_2 < C_3$ for any $x^* \in \mathbb{R}^D$, for any $p$ and $D$.

Unlike our Monte Carlo approximation to the Fisher information matrix, we do not assume a particular type of estimator for $\nabla_\phi \psi(\phi)$. To prove convergence of MC-EIF to the true EIF, we assume that $\nabla_\phi \hat{\psi}_M(\phi)$ converges to $\nabla_\phi \psi(\phi)$ at the following rate:

**Assumption 3.7.** Let $\delta_M := \nabla_\phi \psi(\phi) - \nabla_\phi \hat{\psi}_M(\phi) \in \mathbb{R}^{L \times p}$ denote the approximation error. There exists a universal constant $C_\psi < \infty$ such that for $M > C_\psi$ for any $\epsilon > 0$,

$$\mathbb{P}\left( \|\delta_M\|_F > \sqrt{\frac{p \log p + \epsilon}{M}} \right) < \exp(\text{-}\epsilon), \quad \text{and} \quad \mathbb{P}\left( \|\nabla_\phi \hat{\psi}_M(\phi)\|_F > C_2 \right) < \exp(\text{-}\epsilon),$$

In Appendix A.1, we prove that Monte Carlo estimators of $\nabla_\phi \psi(\phi)$ with gradient clipping [ZHSJ20] satisfy Assumption 3.7. Hence, Assumption 3.7 is a mild condition. Under these three assumptions, and the ones in Theorem 3.4, we prove the following result in Appendix A.1, which states that $M$ must scale linearly with $p \log p$ to guarantee close pointwise approximation.

**Theorem 3.8.** *Suppose Assumption 3.1, Assumption 3.2, Assumption 3.3, Assumption 3.5, Assumption 3.6, and Assumption 3.7 hold. Then, there exists universal constants $0 < C_4$ and $C_5 < \infty$, such that for any $\epsilon > 0$ and $M > \max(C_5(p + \epsilon)C_1^2, C_\psi)$,*

$$|\varphi_\phi(x^*) - \hat{\varphi}_{\phi,M}(x^*)| \leq C_4 \lambda_{max}(\Sigma^{\text{-}1}) \sqrt{\frac{p \log p + \epsilon}{M}}, \tag{5}$$

*for $x^* \in \mathbb{R}^D$ with probability at least $1 - 2\exp(-\epsilon)$, where $\Sigma := cov(\tilde{x})$ and $\lambda_{max}(\cdot)$ denotes the largest eigenvalue of a matrix.*

# 4  MC-EIF for Automated Efficient Inference

In Theorem 3.8, we proved that MC-EIF is close to the true efficient influence function pointwise. In this Section we; (i) show how MC-EIF can be used to automate the construction of popular efficient estimators, and (ii) prove how many Monte Carlo samples are needed to ensure that key statistical properties hold when MC-EIF is used instead of the true EIF. In doing so, MC-EIF brings conceptual clarity to the practice of constructing efficient estimators, and how these estimators can be implemented using existing differentiable probabilistic programming languages like Pyro [BCJ+19].

All three of the efficient estimator templates we explore in this Section involve some combination of plug-in estimation and EIF-based computations. A key practical benefit of our work is that MC-EIF-based efficient estimators are entirely modular; advances in general-purpose probabilistic inference technology directly translate to advances in efficient estimation under our framework.

## 4.1  Von Mises One Step Estimator

We start with the simple *Von Mises One Step Estimator*, which corrects the plug-in estimator in Section 2 by adding the average value of the efficient influence function on a held out dataset. Despite its simplicity, this estimator achieves optimal statistical rates [Ken22]. Our one step estimator using MC-EIF ($\hat{\varphi}_{\phi,M}(x)$) instead of the true efficient influence function ($\varphi_\phi(x)$) is provided in Algorithm 1.

---

**Algorithm 1** MC-EIF one step estimator

---

**Input:** Target functional $\psi$, initial estimate of parameters $\hat{\phi}$, held out datapoints $\{x_n\}_{n=N/2+1}^N$, Number of Monte Carlo samples $M$

$\hat{\theta}_{\text{plug-in}} \leftarrow \psi(\hat{\phi})$          {plug-in estimate}

$C = \frac{2}{N} \sum_{n=N/2+1}^N \hat{\varphi}_{\hat{\phi},M}(x_n)$          {MC-EIF one step correction}

**Return:** $\hat{\theta}_{\text{plug-in}} + C$

---

**Theoretical Guarantees.** We call the one step estimator that uses the true EIF instead of MC-EIF in Algorithm 1 the *analytic one step estimator*. Below we prove how many MC samples are needed to ensure our estimator for finite $M$ has the same statistical properties as the analytic one step estimator.

**Proposition 4.1.** *Let $\hat{\theta}_*$ denote the output of the analytic one step estimator and $\hat{\theta}$ the output of Algorithm 1 for $M = \infty$ and $M < \infty$, respectively. If $M = \Omega(Np \log p)$, $p > O(\log N)$ and the assumptions in Theorem 3.8 hold, then $\|\hat{\theta}_* - \hat{\theta}\|_2 = o_p(1/\sqrt{N})$.*

By Proposition 4.1, MC-EIF is asymptotically efficient when the number of Monte Carlo samples in Algorithm 1 grows faster than $Np \log p$.

## 4.2 Debiased/Double ML

Next, we express debiased/double ML (DML) [CCD+18] in terms of MC-EIF. To rewrite DML explicitly in terms of MC-EIF, we largely follow [CCD+18, IN22].

---

**Algorithm 2** MC-EIF debiased ML

---

**Input:** Vector of estimating equations $g$, initial estimate of parameters $\hat{\phi}$, held out datapoints $\{x_n\}_{n=N/2+1}^N$, Number of Monte Carlo samples $M$

$f(\theta) \leftarrow \frac{2}{N} \sum_{n=N/2+1}^N g(x_n, \eta(p_{\hat{\phi}}), \theta) + \hat{\varphi}_{\hat{\phi},M}(x_n, \theta)$     {MC-EIF orthogonal moment function}

**Return:** $\{\theta : f(\theta) = 0\}$

---

**Construction of Orthogonal Generalized Method of Moment (GMM) Estimators.** GMM-based estimators are defined by a nuisance functional $\eta(\cdot) \in \mathbb{R}^J$, $J \in \mathbb{N}$, and a set of $K \in \mathbb{N}$ functions $\{g_k(x, \eta(\mathbb{P}_\phi), \theta)\}_{k=1}^K$, often called *estimating equations*. These estimating equations are selected so that their roots uniquely identify $\theta^*$ when the nuisance parameters $\eta(\mathbb{P}_\phi)$ are estimated correctly:

$$\mathbb{E}_{x \sim \mathbb{P}_{\phi^*}(x)}[g(x, \eta(\mathbb{P}_{\phi^*}), \theta)] = 0 \iff \theta = \theta^*, \tag{6}$$

where $g := (g_1, \cdots, g_K)$. As an example, $g$ might be the gradient of the log-likelihood function. To make GMM-based estimators less sensitive to incorrect estimation of the nuisance parameters, [CCD+18, IN22, CNS22] replace $g(\cdot)$ with the *orthogonal moment function*, constructed using influence functions. In our setting[3], the orthogonal moment function equals the following:

$$g(x, \eta(\mathbb{P}_\phi), \theta) + \varphi_\phi(x, \theta), \tag{7}$$

where $\varphi_\phi(x, \theta)$ is the efficient influence function associated with the functional $\mu_\theta(\phi) = \mathbb{E}_{x \sim \mathbb{P}_\phi}[g(x, \eta(\mathbb{P}_\phi), \theta)]$ for fixed $\theta$ by Equation 2.6 in [IN22]. By Theorem 3.4,

$$\varphi_\phi(x, \theta) := [\nabla_\phi \mu_\theta(\phi)]^T I(\phi)^{-1} \nabla_\phi \log \mathbb{P}_\phi(x). \tag{8}$$

Since $g$ is a known by assumption, we can readily use the Monte Carlo methods in [KW14, SHWA15] to automatically approximate $\nabla_\phi \mu_\theta(\phi)$. We summarize the DML algorithm in Algorithm 2 which replaces Equation 8 with our MC-EIF approximation.

**Theoretical Guarantees.** For general estimating equations, it is difficult to quantity how errors in our MC-EIF approximation to Equation 8 lead to changes in final estimates. When the estimating equations have more structure, however, we obtain a similar result as in Proposition 4.1.

---

[3]DML can handle functionals which are not pathwise differentiable. As we only consider parametric model families, however, we can assume without loss of generality that the functional is pathwise differentiable [FS23] and that the range space of the nuisance functional is finite-dimensional. Consequently, we can express DML in terms of efficient influence functions.

**Assumption 4.2.** $g(x_n, \eta(\mathbb{P}_\phi), \theta) = m(x_n, \eta(\mathbb{P}_\phi)) - \theta$ for some vector of known functions $m(\cdot)$.

Assumption 4.2 was made in several works [CNS22, IN22] already. We prove an analogous rate guarantee as in Proposition 4.1 under Assumption 4.2.

**Proposition 4.3.** *Let $\hat{\theta}_*$ denote the output of the analytic DML estimator and $\hat{\theta}$ the output of Algorithm 2 for $M = \infty$ and $M < \infty$, respectively. If $M = \Omega(Np \log p)$, $p > O(\log N)$ and the assumptions in Theorem 3.8and Assumption 4.2 hold, then $\|\hat{\theta}_* - \hat{\theta}\|_2 = o_p(1/\sqrt{N})$.*

### 4.3 Targeted Minimum Loss Estimation

We conclude by writing targeted minimum loss estimation (TMLE) [VDLR06] explicitly in terms of MC-EIF. Unlike the one step estimator or DML, TMLE directly corrects the estimated distribution $p_{\hat{\phi}}(x)$ and then plugs in the corrected distribution into the functional $\Psi$ as the final estimate. To perform this correction it perturbs $p_{\hat{\phi}}$ in the direction of the influence function, searching for the optimal step size by maximizing the perturbed likelihood on the held out dataset. Intuitively, TMLE can be viewed as a form of gradient ascent in function space. We show one step TMLE [VDLR06] in Algorithm 3. The multi-step TMLE version is computed by iterating Algorithm 3 multiple times until $\epsilon$ approximately equals 0 [VDLR06].

---

**Algorithm 3** MC-EIF one step TMLE

---

**Input:** Target functional $\Psi$, initial estimate of parameters $\hat{\phi}$, held out datapoints $\{x_n\}_{n=N/2+1}^N$, Number of Monte Carlo samples $M$
$p(\epsilon, x) \leftarrow (1 + \epsilon^T \hat{\varphi}_{\hat{\phi}, M}(x)) p_{\hat{\phi}}(x)$         {MC-EIF projected $\epsilon$-perturbed density function}
$\hat{\epsilon} \leftarrow \arg\max_{\epsilon \in \mathbb{R}^L : p(\epsilon, x) \in \mathcal{M}} \frac{2}{N} \sum_{n=N/2+1}^N \log p(\epsilon, x_n)$     {Maximum likelihood search over $\epsilon$}
**Return:** $\Psi(p(\hat{\epsilon}, \cdot))$

---

## 5 Experiments

We start by comparing the quality of MC-EIF against other methods for influence function approximation. Then, we show how MC-EIF behaves when; (i) the number of Monte Carlo samples is varied, (ii) the dimensionality of the input is varied, and (iii) the efficient estimator type is varied. Our empirical results ultimately validate our theoretical results in Section 3 and Section 4. Finally, we show how MC-EIF can be used to automate the construction of efficient estimators for new functionals by revisiting a classic problem in optimal portfolio theory. Our MC-EIF implementation is publicly available in the Python package ChiRho. All results shown here are end-to-end reproducible.

In [JWZ22a], the authors target the nonparametric influence function, which is the unique influence function when $\mathcal{M} = L^2(\mathbb{P})$ in Definition 2.1. By contrast, we target the efficient influence function. Thus, for evaluation, we compare how well the empirical Gateaux method from [JWZ22a] approximates the nonparametric influence function and how well our MC-EIF method approximates the efficient influence function on the same data-generating process.

To have a ground truth for comparison, we select a simple model and functional where we can analytically compute the nonparametric and efficient influence functions. To this end, we consider the problem of estimating the expected density, $\Psi(\mathbb{P}) = \int \mathbb{P}(x)^2 dx$ as in [BR88, CLvdL19]. We further suppose that $x \sim N(\mu, \sigma)$. We consider two parametric model families: one where $\mu$ is unknown but $\sigma = 1$, and one where both $\mu$ and $\sigma$ are unknown, which we call $\mathcal{M}_1$ and $\mathcal{M}_2$ respectively. As the nonparametric influence function makes no assumptions on the underlying model family, it remains fixed across $\mathcal{M}_1$ and $\mathcal{M}_2$ and always equals $2(\mathbb{P}(X) - \Psi(\mathbb{P}))$ [CLvdL19]. However, the EIF equals zero for $\mathcal{M}_1$, as the expected density does not depend on $\mu$. Hence, any plug-in estimate for models in $\mathcal{M}_1$ will result in a correct value of the expected density, and thus no distributional perturbations produce any change. In $\mathcal{M}_2$, the efficient influence function for the expected density depends on the unknown $\sigma$. See Figure 7 in the Appendix for further intuition around the expected density influence functions in parametric (in unknown $\sigma$) and non-parametric settings.

Figure 1 summarizes how well the empirical Gateaux derivative method approximates the nonparametric influence function and how well our MC-EIF method approximates the EIF at the point

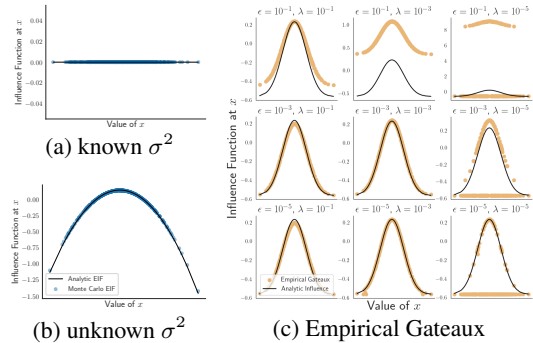

(a) known $\sigma^2$

(b) unknown $\sigma^2$

(c) Empirical Gateaux

Figure 1: **Comparison between MC-EIF and empirical Gateaux approximation.** MC-EIF (a and b) is less sensitive to hyperparameters ($\epsilon$ and $\lambda$) than the empirical Gateaux baseline (c).

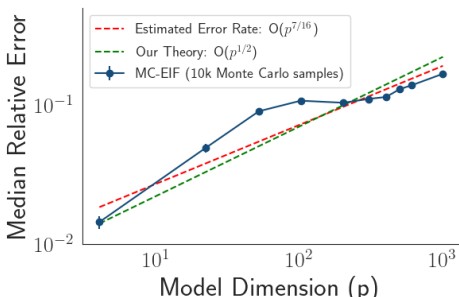

Figure 2: **Empirical evidence for convergence theory.** Increasing $p$ for the ATE experiments produces MC-EIF approximation errors that closely match and sit below the worst-case error rates given by Theorem 3.8.

$\mathbb{P}_\phi = N(0, 1)$. We see that MC-EIF is able to approximate the efficient influence function very well ($M = 10^4$ samples). By contrast, the empirical Gateaux derivative is highly sensitive to the choice of two kernel smoothing hyperparameters, $\epsilon$ and $\lambda$. As the true influence function is not known, it is not always clear how to select $\epsilon$ and $\lambda$.[4] Such numerical instability was already discussed in [CLvdL19], where the precision necessary must get exponentially smaller with input dimension, making it infeasible when $D \approx 10$.[5] MC-EIF, however, has only a single tunable parameter ($M$), where larger $M$ unambiguously provides a better approximation. In Theorem 3.8, we provided conditions for this improvement, and Figure 9 of the Appendix corroborates the unsurprising improvement empirically. We further discuss challenges in automating the empirical Gateaux method in Appendix C. We attempted to use the empirical Gateaux derivative as a baseline for other experiments, but were unable to achieve numerically stable solutions for any $p > 2$ without prohibitively long run-times.

Next, we focus on a classic model consisting of a binary treatment, high-dimensional continuous confounders, and Gaussian distributed response; see Appendix E for the precise model formula. We assume that the analyst is interested in estimating the average treatment effect (ATE), where the true ATE is zero but unknown. All influence function computations are relative to an initial point estimate $\hat{\phi}$, found through maximum a posteriori estimation using 500 training datapoints. Due to the exponential runtime in dimension for the methods in [JWZ22a, CLvdL19], we focus on comparing MC-EIF with the analytic influence function for ATE below.

**Sensitivity to Dimensionality.** Theorem 3.8 implies that for a fixed number of Monte Carlo samples $M$, the quality of the approximation degrades with the square root of model dimension $p$. In Figure 2, we empirically show how approximation quality degrades as $p$ increases for $M = 10^4$ fixed. Based on Figure 2, the empirical results closely match the theoretical behavior predicted by Theorem 3.8.[6] We also show how the computational complexity of MC-EIF scales as $p$ increases in Figure 2.

**Sensitivity to Estimator Type.** Here, we consider a high-dimensional setup where there are 200 confounders but only 500 training datapoints. We simulate 100 different datasets with this configuration to approximate the sampling distribution of different efficient estimators. In Figure 3, we see that across estimators, using MC-EIF instead of the true EIF results in minimal downstream error. This is consistent with our theoretical results in Section 4. While MC-EIF is agnostic to the choice of

---

[4]One suggestion is to visually inspect an epsilon-lambda plot (like the one in Figure 1) for a "...possibly curvilinear triangular region nested in the upper left portion of the [plot]. In this region, the finite-difference approximation of the EIF value should be essentially constant ([CLvdL19], §4.3)."

[5]Specifically, where the non-parametric dimensionality of the data unit is $d = d_1 + d_2$, current theory requires $\epsilon \ll \lambda^{d_1}$, or even $\epsilon \ll \lambda^d$ in some largely avoidable cases ([CLvdL19], §4.1). If functional evaluation is approximate, smaller $\epsilon$ requires significantly more compute for the finite difference to dominate Monte Carlo error in estimating the difference between the functional evaluated on the plugin distribution, and the functional evaluated on the $\epsilon$-perturbed distribution. $\epsilon$ might even be so small as to overrun floating point accuracy on modern machines — even $.1^{16}$ exceeds standard precision recommendations for 64-bit floating point values.

[6]As discussed in Section 3.2, to make the error not grow with $p$, we would need $M \asymp p \log p$.

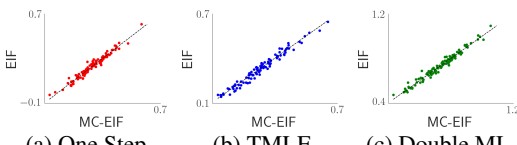

| (a) One Step | (b) TMLE | (c) Double ML |
|---|---|---|

Figure 3: **Comparison between ATE estimators using MC-EIF and analytic EIF.** MC-EIF produces ATE estimates very close to the diagonal, representing an oracle estimator of the EIF.

| Metric | One Step MC-EIF | Plug-in |
|---|---|---|
| REV | $\mathbf{1.86 \pm .35}$ | $2.60 \pm .35$ |
| RMSE | $\mathbf{.08 \pm .02}$ | $.14 \pm .02$ |

Table 1: **Empirical results for Markowitz optimal portfolio optimization.** Using MC-EIF, Algorithm 1 achieves lower relative expected volatility (REV) and RMSE compared to the oracle estimator.

efficient estimator, one may prefer some over others depending on the problem. See Figures 5 and 6 of the Appendix for an example performance comparison between efficient estimators of the ATE.

**Ability to Handle New Functionals.** To illustrate MC-EIF's flexibility, we revisit a classic problem in optimal portfolio theory. Suppose that $x \in \mathbb{R}^D$ is a vector of asset returns. We are interested in estimating the optimal portfolio weights $\theta^* \in \mathbb{R}^D$ that maximize the expected return while minimizing the variance of the portfolio. Then, the Markowitz optimal portfolio [Mar52] is given by:

$$\Psi_\lambda(\mathbb{P}_\phi) = \arg \max_{\theta \in \mathbb{R}^D} \theta^T \mathbb{E}_{\mathbb{P}_\phi}[x] - \lambda \theta^T \mathrm{Cov}(x; \mathbb{P}_\phi)\theta, \qquad \text{subject to } \sum_{i=1}^D \theta_i = 1, \qquad (9)$$

where $\lambda$ is the tradeoff between expected return and variance (measure of risk), and $\mathrm{Cov}(x; \mathbb{P}_\phi)$ denotes the covariance matrix with respect to $\mathbb{P}_\phi$. Hence, the optimal weights functional $\Psi_\lambda(\mathbb{P}_\phi)$ depends on a high-dimensional nuisance, namely the $D \times D$ covariance matrix of returns. The target $\theta^*_{\phi,\lambda} = \Psi_\lambda(\mathbb{P}_\phi)$ is a much lower $D$-dimensional target parameter. Setting $\lambda = \infty$ corresponds to the *global minimum variance portfolio* [HB91, JM03, ARU20], for which there is (to our knowledge) no efficient estimator in the literature. We show results in Table 1 indicating substantial improvement in a synthetic data experiment; a detailed description of this experiment may be found in Appendix E.2.

## 6 Limitations

As discussed in Assumptions 3.1 and 3.2, both the likelihood and the target functional must be differentiable with respect to $\phi$. In practice, especially if the model involves latent discrete random variables, some degree of relaxation, marginalization, or reparameterization may be required to ensure differentiability [JGP17]. Recall also that while MC-EIF operates on models with finite parametrizations (Section 3), its capacity to handle high-dimensional nuisance parameters means it can likely apply to, for example, function approximators that recover some of the value proposition offered by non-parametric model components [HSW89]. Additionally, as discussed in Appendix D, infinite-dimensional models (like the Gaussian process) can often be reduced to finite ones where MC-EIF can be applied. That said, future work is needed to fully explore the practical and empirical capabilities of MC-EIF in these settings, including how the polynomial complexity of Fisher information matrix inversions plays out in practice.

## 7 Conclusion

We have shown both theoretically and empirically that MC-EIF can reliably be used to automate efficient estimation. Our key contributions include MC-EIF's consistency and capability to achieve optimal convergence rates. Empirical evidence shows that MC-EIF performs comparably to traditional estimators using analytic EIFs. Additionally, we illustrate the practical application of MC-EIF in scenarios where the analytic EIF is not known. Given these contributions, there are many exciting areas of future work. For example, one may with to construct more powerful provably efficient estimators on top of MC-EIF (see Appendix D) and explore the growing connection between semi-parametric theory and heuristic methods in deep learning [VACB22, BNL$^+$22, DKSM21, ZDJ$^+$23]. Additionally, there are many methods that could be used to accelerate the calculation of the Fisher information matrix, which is a computational bottleneck in MC-EIF. Given its foundational role in statistics, various techniques—such as using Kronecker-factored approximations [GM16]—could improve efficiency without sacrificing performance.

## Acknowledgments and Disclosure of Funding

The authors would like to thank DARPA for funding this work through the Automating Scientific Knowledge Extraction and Modeling (ASKEM) program, Agreement No. HR0011262087. The views, opinions and/or findings expressed are those of the authors and should not be interpreted as representing the official views or policies of the Department of Defense or the U.S. Government. The authors would also like to thank Tamara Broderick, David Burt, and Ryan Giordano for helpful discussions.

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

## A  Proofs

### A.1  Proof of Theorem 3.8

*Proof.* Let $\{x_m\}_{m=1}^M$ denote the $M$ Monte Carlo samples in Equation 3, and let $\tilde{x}_m :=$ $\frac{1}{\sqrt{D}}\nabla_\phi \log \mathbb{P}_\phi(x_m)$ for $1 \le m \le M$. Let $\hat{\Sigma} = \frac{1}{M}\sum_{m=1}^M \tilde{x}_m \tilde{x}_m^T$ denote the sample covariance matrix. Then, $\Sigma = \frac{1}{D}I(\phi)$ and $\hat{\Sigma} = \frac{1}{D}\hat{I}(\phi)$. Hence,

$$
\begin{aligned}
|\varphi_\phi(x) - \hat{\varphi}_{\phi,M}(x)| &= \left| [\nabla_\phi \hat{\psi}_M(\phi)]^T (\Sigma^{-1} - \hat{\Sigma}^{-1})\frac{\nabla_\phi \log \mathbb{P}_\phi(x^*)}{D} + \delta_M \Sigma^{-1}\frac{\nabla_\phi \log \mathbb{P}_\phi(x^*)}{D} \right| \\
&\le \left| [\nabla_\phi \hat{\psi}_M(\phi)]^T (\Sigma^{-1} - \hat{\Sigma}^{-1})\frac{\nabla_\phi \log \mathbb{P}_\phi(x^*)}{D} \right| + \left| \delta_M \Sigma^{-1}\frac{\nabla_\phi \log \mathbb{P}_\phi(x^*)}{D} \right| \\
&\le \left| [\nabla_\phi \hat{\psi}_M(\phi)]^T (\Sigma^{-1} - \hat{\Sigma}^{-1})\frac{\nabla_\phi \log \mathbb{P}_\phi(x^*)}{D} \right| + \left| \delta_M \Sigma^{-1} C_3 \right| \\
&\le \left| [\nabla_\phi \hat{\psi}_M(\phi)]^T (\Sigma^{-1} - \hat{\Sigma}^{-1})\frac{\nabla_\phi \log \mathbb{P}_\phi(x^*)}{D} \right| + C_3 \lambda_{\max}(\Sigma^{-1})\|\delta_M\|_F
\end{aligned}
\tag{10}
$$

By Assumption 3.7, $\|\delta_M\|_F < \sqrt{\frac{p+\epsilon}{M}}$ with probability greater than $1 - \exp(-\epsilon)$ when $M > C_\psi$. If we can prove that there exists a constant $C_4$

$$
\left| [\nabla_\phi \hat{\psi}_M(\phi)]^T (\Sigma^{-1} - \hat{\Sigma}^{-1})\frac{\nabla_\phi \log \mathbb{P}_\phi(x^*)}{D} \right| \le C_4 \lambda_{\max}(\Sigma^{-1})\sqrt{\frac{p\log p + \epsilon}{M}}
\tag{11}
$$

with probability greater than $1 - 2\exp(-\epsilon)$, the claim follows by an application of the union bound. By Theorem 10 in [KK21], the claim follows if we can prove that there exists a universal constant $C_4$ such that

$$
\left\| \frac{\nabla_\phi \log \mathbb{P}_\phi(x^*)}{D}^T \Sigma^{-1}\tilde{x} \right\|_{\psi_2} \left\| \nabla_\phi \hat{\psi}_M(\phi)^T \Sigma^{-1}\tilde{x} \right\|_{\psi_2} < C_4 \lambda_{\max}(\Sigma^{-1}),
\tag{12}
$$

with probability greater than $1 - \exp(-\epsilon)$, where $\|\cdot\|_{\psi_2}$ denotes the Orlicz sub-Gaussian norm; see Equation 9 in [KK21] for a precise definition of the Orlicz norm of a random vector. With probability greater than $1 - \exp(-\epsilon)$, $\|\nabla_\phi \hat{\psi}_M(\phi)\|_F < C_2$ by Assumption 3.7. Hence, with probability greater than $1 - \exp(-\epsilon)$,

$$
\begin{aligned}
\left\| \frac{\nabla_\phi \log \mathbb{P}_\phi(x^*)^T}{D} \Sigma^{-1} \tilde{x} \right\|_{\psi_2} \left\| \nabla_\phi \hat{\psi}_M^T \Sigma^{-1} \tilde{x} \right\|_{\psi_2} &\leq C_2 C_3 \left\| \Sigma^{-1} \tilde{x} \right\|_{\psi_2}^2 \\
&\leq C_2 C_3 \| \Sigma^{-1/2} \|_2^2 \| \Sigma^{-1/2} \tilde{x} \|_{\psi_2}^2 \\
&\leq C_1 C_2 C_3 \| \Sigma^{-1/2} \|_2^2 \| \mathrm{cov}(\Sigma^{-1/2} \tilde{x}) \|_2 \\
&= C_1 C_2 C_3 \| \Sigma^{-1/2} \|_2^2 \\
&= C_1 C_2 C_3 \lambda_{\max}(\Sigma^{-1}),
\end{aligned}
\tag{13}
$$

where the first inequality follows from Assumption 3.6, the third by [DJS08] and Assumption 3.5, and last by the definition of the spectral norm of a matrix. The result now follows by setting $C_4 = C_1 C_2 C_3$. $\qquad\square$

**Assumption 3.7 Holds for Monte Carlo Estimators.** Here we show if $\hat{\psi}_M$ is also approximated with $M$ Monte Carlo samples, then Assumption 3.7 holds. To this end, suppose

$$
\nabla_\phi \hat{\psi}_M(\phi) = \frac{1}{M} \sum_{m=1}^{M} \nabla_\phi g_\phi(w_m), \quad w_m \overset{\text{iid}}{\sim} q(w) \quad 1 \leq m \leq M, \quad \text{s.t.} \quad \mathbb{E}[\nabla_\phi g_\phi(w_m)] = \nabla_\phi \psi(\phi),
\tag{14}
$$

for some distribution $q(w)$ and function $g_\phi$. Such a decomposition exists, for example, when the functional is expressible as a stochastic computation graph [SHWA15] or for reparameterizable densities [KW14]. Suppose further that there exists a universal constant such that $\nabla_{\phi_j} g_\phi(w_m) \in \mathbb{R}^L$ is a sub-Gaussian random vector with parameter $\sigma_\psi$ for $1 \leq j \leq p$. Then,

$$
\begin{aligned}
\|\delta_M\|_F &= \sqrt{ \sum_{j=1}^{p} \sum_{l=1}^{L} ([(\nabla_{\phi_j} g_\phi(w_m)]_l - [(\nabla_{\phi_j} \psi(w_m)]_l)^2 } \\
&\leq \sqrt{pL} \max_{1 \leq l \leq L, 1 \leq j \leq p} |[(\nabla_{\phi_j} g_\phi(w_m)]_l - [(\nabla_{\phi_j} \psi(w_m)]_l|
\end{aligned}
\tag{15}
$$

By Exercise 2.12 in [Wai19],

$$
\max_{1 \leq l \leq L, 1 \leq j \leq p} |[(\nabla_{\phi_j} g_\phi(w_m)]_l - [(\nabla_{\phi_j} \psi(w_m)]_l| = O_p\left( \sigma_\psi \sqrt{\frac{\log(pL)}{M}} \right)
\tag{16}
$$

Hence, since $L$ is a constant, $\|\delta_M\|_F = O_p\left( \sqrt{\frac{p \log(p)}{M}} \right)$. Thus, the first equation in Assumption 3.7 holds. Under Assumption 3.6, the second equation in Assumption 3.7 trivially holds using gradient clipping with $C_2$.

### A.2 Proof of Theorem 4.1

*Proof.* We want to prove that difference between the analytic one step estimator and Algorithm 1 with finite $M$ decays at a $o_p\left(\frac{1}{\sqrt{N}}\right)$ rate. Their difference equals

$$
\left| \frac{2}{N} \sum_{n=N/2+1}^{N} \left( \varphi_{\hat{\phi}}(x_n) - \hat{\varphi}_{\hat{\phi},M}(x_n) \right) \right| \leq \max_{n=N/2+1,\cdots,N} \left| \varphi_{\hat{\phi}}(x_n) - \hat{\varphi}_{\hat{\phi},M}(x_n) \right|.
\tag{17}
$$

Let $\epsilon > 0$ and suppose $M' = C_4^2 \max(C_5, 1)(p \log(p) + \epsilon) \max(C_1^2, 1) N \lambda_{\max}^2(\Sigma^{-1}) = O(p \log p N)$, where $C_1$ is defined in Assumption 3.5, and $C_5$ and $\Sigma^{-1}$ are defined in Theorem 3.8. If

$$
\mathbb{P}^* \left( \max_{n=N/2+1,\cdots,N} \left| \varphi_{\hat{\phi}}(x_n) - \hat{\varphi}_{\hat{\phi},M'}(x_n) \right| > \sqrt{\frac{2}{N}} \right) \leq \exp(-\epsilon).
\tag{18}
$$

holds, then the proof is complete since $M$ grows faster than $M'$. Let $\epsilon_N = \epsilon + \log(N/2)$. By Theorem 3.8 and the union bound,

$$
\mathbb{P}^* \left( \max_{n=N/2+1,\cdots,N} \left| \varphi_{\hat{\phi}}(x_n) - \hat{\varphi}_{\hat{\phi},M'}(x_n) \right| > C_4 \lambda_{\max}(\Sigma^{\text{-}1}) \sqrt{\frac{p\log(p) + \epsilon_N}{M'}} \right)
$$
$$
\leq \frac{N}{2} \sum_{n=N/2+1}^{N} \mathbb{P}^* \left( \left| \varphi_{\hat{\phi}}(x_n) - \hat{\varphi}_{\hat{\phi},M'}(x_n) \right| > C_4 \lambda_{\max}(\Sigma^{\text{-}1}) \sqrt{\frac{p\log(p) + \epsilon_N}{M'}} \right) \quad (19)
$$
$$
\leq N/2 \exp(-\epsilon_N)
$$
$$
= \exp(-\epsilon_N + \log N/2)
$$
$$
= \exp(-\epsilon)
$$

Now,

$$
C_4 \lambda_{\max}(\Sigma^{\text{-}1}) \sqrt{\frac{p\log(p) + \epsilon_N}{M'}} = C_4 \lambda_{\max}(\Sigma^{\text{-}1}) \sqrt{\frac{p\log(p) + \epsilon + \log(N/2)}{C_4^2 \max(C_5, 1)(p + \epsilon)\max(C_1^2, 1)N\lambda_{\max}^2(\Sigma^{\text{-}1})}}
$$
$$
\leq \sqrt{\frac{p\log(p) + \epsilon + \log(N/2)}{(p\log(p) + \epsilon)N}}
$$
$$
\leq \sqrt{\frac{2}{N}}. 
$$
$$(20)$$

The proof now follows from Equation 19 and Equation 20.

$\square$

## A.3 Proof of Theorem 4.3

**Lemma A.1.** *Suppose Assumption 4.2 holds. Then, $\hat{\theta}_{DML} = \frac{2}{N}\sum_{n=N/2+1}^{N} m(x_n, \eta(p_{\hat{\phi}})) + \sum_{n=1}^{N} \varphi_{\hat{\phi}}(x_n)$, where $\varphi_{\hat{\phi}}(x)$ is the influence function associated with the functional $\psi(\phi) = \mathbb{E}_{x \sim \mathbb{P}_\phi(x)}[m(x_n, \eta(\mathbb{P}_\phi))]$.*

*Proof.* We claim $\varphi_\phi(x, \theta) = \varphi_\phi(x)$ for all $\theta$. To prove this claim, notice that $\nabla_\phi \mu_\theta(\phi) = \nabla_\phi \mu_{\theta'}(\phi)$ for arbitrary $\theta$ and $\theta'$ since $\mu_\theta(\phi) = \mathbb{E}_{x \sim \mathbb{P}_\phi}[m(x_n, \eta(\mathbb{P}_\phi))] - \theta$. Hence, the claim follows from Equation 8. Now,

$$
\frac{2}{N} \sum_{n=N/2+1}^{N} \left[ g(x_n, \eta(p_{\hat{\phi}}), \theta) + \varphi_{\hat{\phi}}(x_n, \theta) \right] = \frac{1}{N} \sum_{n=1}^{N} \left[ m(x_n, \eta(p_{\hat{\phi}})) + \varphi_{\hat{\phi}}(x_n) \right] - \theta. \quad (21)
$$

Hence, $\hat{\theta}_{\text{DML}} = \frac{2}{N}\sum_{n=N/2+1}^{N} m(x_n, \eta(p_{\hat{\phi}})) + \sum_{n=1}^{N} \varphi_{\hat{\phi}}(x_n)$. $\square$

*Proof of Proposition 4.3.* By Lemma A.1, Algorithm 2 uses the same correction term $C$ in Algorithm 1. Hence, the proof of Proposition 4.3 now follows from Proposition 4.1. $\square$

*Remark* A.2. By Lemma A.1, the only difference between Algorithm 2 and Algorithm 1 is a different value for the initial estimate of $\theta^*$. Specifically, in DML, the initial estimate of $\theta^*$ is $\frac{2}{N}\sum_{n=N/2+1}^{N} m(x_n, \eta(p_{\hat{\phi}}))$, which averages over datapoints drawn from the true distribution. By contrast, Algorithm 1 uses $\hat{\theta}_{\text{plug-in}}$, which averages over datapoints *simulated* from $p_{\hat{\phi}}$.

# B Code Examples

## B.1 Automatically Differentiable Functionals

The implementation of differentiable functional approximations is fairly straightforward when using modern autodifferentation tools. For example, the squared density functional for a mean-zero, univariate normal can be approximated using Monte Carlo as follows:

$$\frac{1}{N} \sum_{n=1}^{N} \mathcal{N}\left(x_n, \sigma^2\right); \;\; x_n \sim \mathcal{N}\left(0, \sigma^2\right) \tag{22}$$

This can be implemented in pytorch [PGC$^+$17], and thus automatically differentiated with respect to $\sigma$ (called "scale" in the code block below) using, for example, `torch.autograd.grad`.

Listing 1: Automatically Differentiable Monte Carlo Approximation of Integrated Squared Normal Density

```
import torch

def diffable_mc_integ_squared_norm_density(scale: torch.Tensor, num_monte_carlo: int):
    assert scale.requires_grad
    # Sample from the density
    samples = torch.distributions.Normal(0., scale).sample((num_monte_carlo,))
    # Evaluate those samples under the density.
    logprobs = torch.distributions.Normal(0., scale).log_prob(samples)
    # Return the mean pdf value in a numerically stable way.
    return torch.exp(torch.logsumexp(logprobs, dim=0)) / torch.numel(samples)
```

## C  So, What's Automatic?

| | | Derivation of IF (or approximation) | Considerations for Numerical Approximation | Implementation in Computer Code |
|---|---|---|---|---|
| Non-Parametric IF | Analytic | Manual | Often Straightforward | Manual |
| | Empirical Gauteaux | Straightforward | Manual | Manual |
| Efficient IF | Ours requires $\nabla_\phi \hat{\psi}$ | Automated | Automated | Automated |

Figure 4: We taxonomize the workflow of robust estimation into three stages: the derivation of an (approximate and/or efficient) influence function, the numerical derivation and analysis required for its computation, and the code required to compute it. For the analytic workflow, the derivation of the IF results in Equation 24. This largely involves terms already required by the original plug-in (Equation 23), but still must be implemented on a case-by-case basis in code. For the "Empirical Gateaux" workflow, the first stage requires only the general purpose Equation 25, but demands case-specific numerical considerations and derivations like the one shown in Equation 26. In stark contrast, given a differentiable approximation to the functional of interest, MC-EIF "automates" each stage through use of an end-to-end, general purpose solution.

The work required to perform robust estimation can be subdivided into a few key steps. The process begins with a functional of interest, $\Psi$. With this functional in hand, an analyst must first derive the influence function (or an approximation thereof), and consider any nuances in numerically approximating that quantity. Finally, an engineer must implement that approximation as executable code. Different approaches boast varying levels of "automation" for each step. We claim that in problems where our conditions hold (as outlined in Section 3.2), MC-EIF provides end-to-end automation via a general-purpose solution at each stage. Here, we contrast our approach with both the analytic (see e.g. [Ken16]) and "Empirical Gateaux" workflows [JWZ22b, CLvdL19]. We will track a workflow's "products" at each of the three stages: first, the derivation of the (efficient and/or approximate) influence function; second, a tractable version of the influence function that properly considers its numerical nuances; third, executable code that computes the influence function. Because our approach uses a general purpose formulation for each of these three stages, we call our approach "automated."

We assume the workflow starts having identified a functional of interest and having implemented, in code, a plug-in estimator for it. Throughout, we will use the mean-potential outcome (MPE) functional as our working example[7]:

$$\Psi\left(P\right) = \mathbb{E}_P\left[\mathbb{E}_P\left[Y \mid X, A = 1\right]\right] = \int \int y \frac{p\left(y, A = 1, x\right)}{p\left(A = 1, x\right)} p\left(x\right) dy dx \qquad (23)$$

### C.1  Analytic Workflow

The analytic workflow begins by deriving a closed form influence function — a challenging task even for seasoned experts. This first stage culminates in the following analytic influence function for the MPE [Ken16].

$$\varphi\left(O; P\right) = \frac{\mathbb{I}\left(A = 1\right)}{P\left(A = 1 \mid X\right)} \left\{Y - \mathbb{E}_P\left[Y \mid X, A = 1\right]\right\} + \mathbb{E}_P\left[Y \mid X, A = 1\right] - \Psi\left(P\right) \qquad (24)$$

For general functionals, the derivation resulting in Equation 24 is challenging, even for experts — but given such a derivation, it is often the case that the computation of the quantities composing it can share code and numerical considerations developed to estimate the original, plug-in functional (e.g. Equation 23). Indeed, the influence function of the MPE (Equation 24) involves only terms that an analyst has already considered and implemented for the plug-in (Equation 23). For this reason, we say that in the "analytic" workflow, most of the labor must be allocated to deriving the influence function — tractable, well behaved computation of that influence function tends to involve straightforward extensions of tooling and analysis that already exists for the plug-in.

The last stage is the implementation of that tooling in computer code, which will always require some work on a case-by-case basis.

### C.2  Empirical Gateaux Workflow

The workflow presented by [JWZ22b] and [CLvdL19] significantly reduces the resources required in the first stage — the derivation of the influence function — by providing a general purpose, finite-difference approximation to the influence function (Equation 25). $\tilde{P}_{\epsilon,\lambda}$, here, represents a perturbation of the estimated distribution $\tilde{P}$ of size $\epsilon$ in the direction of a $\lambda$-smooth kernel centered at observation $O$.

$$\tilde{\varphi}\left(O; \tilde{P}\right) = \frac{1}{\epsilon}\left(\Psi\left(\tilde{P}_{\epsilon,\lambda}^O\right) - \Psi\left(\tilde{P}\right)\right) \qquad (25)$$

At first glance, it seems that computing this term would follow easily given a general purpose framework for the perturbation of $\tilde{P}$, and then applying the plug-in functional to that perturbed density. Unfortunately, computing $\Psi\left(\tilde{P}_{\epsilon,\lambda}^O\right)$ presents a number of numerical challenges in practice. As exhibited in Figure 1 (which echoes figure 1 in the work by [CLvdL19]), selecting appropriate perturbation parameters $\epsilon$ and $\lambda$ a priori is challenging, and a battle-tested framework for doing so has not yet been developed. Further, in high dimensions (where MC-EIF excels), the required $\epsilon$ can be so small as to quickly overrun floating point accuracy on modern computers when even $D \approx 10$ [CLvdL19]. Indeed, [JWZ22b] have explicitly left thorough numerical analysis of this approach to further work. In footnote 7, they anecdotally report that quadrature methods were overly sensitive in evaluating perturbed densities in the MPE functional, and instead present a Monte Carlo approach tailored to the task. Unfortunately, neither the numeric considerations or code-implementations of the plug-in estimator easily translate when computing the plug-in with respect to the *perturbed* data distribution. Below, we show their numeric approximation[8] of Equation 25 for the MPE, where observation $o$ comprises $(x, a, y)$, the perturbation kernel $K$ has bandwidth $\lambda$, and they use $N$ Monte Carlo samples from a uniform kernel over confounder $x$.

---

[7]For simplicity in exposition, but without loss of generality in regards to this description of what is and isn't automated, we follow [JWZ22b] in assuming that outcome and confounders are continuous real numbers, while the treatment is binary.

[8]This is their equation 73 in appendix E.3.

$$\tilde{\varphi}_{\lambda,\epsilon}\left(o\right) = \frac{1}{N}\sum_k \left( \frac{(1-\epsilon)\left(\sum_{j:A_j=1} K(X_j - \tilde{x}_k)Y_j\right)P(A=1) + \epsilon y_i \mathbb{I}\left[a_i = 1\right] \cdot 1}{(1-\epsilon)p(A=1,\tilde{x}_k) + \epsilon \mathbb{I}\left[a_i = 1\right]} \right)$$
$$+ (1-\epsilon)\frac{1}{N}\sum_k \frac{\tilde{p}(\tilde{x}_k)}{\tilde{p}_\epsilon(A=1,\tilde{x}_k)} \mathbb{I}\left[a_i = 1\right]\{y_i - \mathbb{E}_{\tilde{P}}[Y|A=1,\tilde{x}_k]\} \tag{26}$$

Indeed, just like finite differencing can simplify multivariate calculus, but introduce numeric challenges, the empirical-gateaux approach makes variational calculus easier, but introduces numeric challenges. In sum, we consider the second, "numerical," stage of this workflow to be both labor and expertise intensive, even when the curse of dimensionality does not render it moot.

Like in the analytic workflow, the "coding" stage of course requires case-by-case implementations. Moreover, added numerical challenges here introduce significant nuance in implementation that isn't present in the analytic case.

### C.3  Our Workflow

In stark contrast, our workflow exploits general solutions in all three stages for the "price" of differentiability of a parametric plug-in estimator. When our general conditions are met (as outlined in Section 3), Equation 1 provides the general purpose solution to the first stage of deriving an (approximate, efficient) influence function, and the second stage is achieved with the Monte Carlo approximation in Equation 2.

The third stage is met in software implementing this general purpose solution that operates on functional implementations using one of many auto-differentiation tools now ubiquitous in machine learning (see Appendix B.1 for a simple example). As discussed at the end of Section 3.1, this sometimes requires the ability to exploit methods like the reparameterization trick for the functional of interest. In many cases, modern automatic differentiation software makes this trivial (as shown in Appendix B.1). For some functionals, however, like those involving inner optimizations, this may be more challenging.

This general purpose approximation underpins the end-to-end automation of MC-EIF, and is to the best of our knowledge the only such general purpose approximation for efficient influence functions.

## D  Towards an EIF Cookbook

As a generalization of the gradient operator on ordinary functions, the EIF viewed as an operator on functionals can be shown to have a number of convenient algebraic properties [Ken16, Ken22], many of which are inherited directly by the MC-EIF estimator. In this section, we speculate on several ways in which these properties could be used to extend the basic MC-EIF framework (and the MC-EIF-based robust estimators in Section 4) to new classes of models and functionals, significantly increasing the range of practical use cases addressable by an implementation of MC-EIF in a differentiable probabilistic programming language like Pyro [BCJ+19].

**Multi-argument functionals** Many important quantities in statistics and machine learning, like the mutual information $\mathbb{I}[X;Y]$ or KL-divergence $\mathbb{KL}[P;Q]$ are functionals of more than one probability distribution. As shown in [KKP+15], we can define partial EIFs analogous to partial derivatives for these quantities (which can then be plugged into the efficient estimators of Section 4) by treating all but one argument as part of the functional and computing the ordinary EIF with respect to that argument.

**Higher-order EIFs** Although all of the efficient estimators of Section 4 are derived from the first-order EIF, there are some circumstances where incorporating higher-order EIFs can be shown to be theoretically necessary for achieving certain statistical properties [RLT+08, BKW23]. Just as ordinary higher-order derivatives are computed by recursively applying a first-order derivative operator to its output, higher-order EIFs can be computed by recursively applying a first-order EIF operator to its own output [vdV14], a property straightforwardly inherited by MC-EIF.

**Models with latent variables** Thus far, we have assumed that we can exactly simulate from model predictive distributions $x \sim p_\phi$ and compute log-densities $\log p_\phi(x)$, score functions $\nabla_\phi \log p_\phi(x)$ and Hessian-vector products. However, our MC-EIF estimator can be extended straightforwardly to models with latent variables and intractable densities and score functions by using a nested Monte Carlo procedure [RCY+18, SW23] to approximate the prior predictive or posterior predictive distributions and plugging the resulting stochastic estimates into the vanilla MC-EIF framework. We expect our theoretical results to extend to this case provided the approximation error can be made small relative to the Monte Carlo error in estimating the Fisher matrix from a finite set of model Monte Carlo samples.

**Infinite-dimensional models and targets** Semiparametric statistics is by definition fundamentally concerned with models that contain infinite-dimensional (i.e. function-valued) components. There is also intense interest in deriving efficient, doubly robust estimators for infinite-dimensional target functionals like the conditional average treatment effect (CATE) in causal inference. Fortunately, in many of these settings the infinite-dimensional quantities can be reduced to finite ones (and ultimately must be to be representable on a digital computer) to which MC-EIF may be applied in a straightforward way. For example, a Gaussian process is fully characterized by the latent function's values on a finite set of test points; computing the EIF for a functional of the GP reduces to computing the EIF of the finite-dimensional joint distribution on function values at the test points, which is straightforward to estimate with MC-EIF. Similarly, in the case of the CATE, we are ultimately interested in the values of the CATE function on a finite set of test inputs, reducing the problem to ordinary MC-EIF for a finite-dimensional target functional.

# E  Additional Experiment Details

## E.1  Model Details

In Section 5, we consider the following model with confounders $c$, treatment $t$, and response $y$:

$$
\begin{aligned}
\mu_0 &\sim \mathcal{N}(0,1), \quad \text{(intercept)} \\
\xi &\sim \mathcal{N}\left(0, \frac{1}{\sqrt{F}} I_F\right), \quad \text{(outcome weights)} \\
\pi &\sim \mathcal{N}\left(0, \frac{1}{\sqrt{F}} I_F\right), \quad \text{(propensity weights)} \\
\tau &\sim \mathcal{N}(0,1), \quad \text{(treatment weight)} \\
c_n &\sim \mathcal{N}(0, I_D), \quad \text{(confounders)} \\
t_n \mid c_n, \pi &\sim \text{Bernoulli}(\text{logits} = \pi^T c_n), \quad \text{(treatment assignment)} \\
y_n &\sim \mathcal{N}(\tau t_n + \xi^T c_n + \mu_0, 1), \quad \text{(response)}
\end{aligned}
\tag{27}
$$

where $F \in \mathbb{N}$ denotes the number of confounders. In this example, $x = (c, t, y) \in \mathbb{R}^D$, where $D = F + 2$ and $\phi = (\mu_0, \xi, \pi, \tau) \in \mathbb{R}^{2F+2}$. To obtain a point estimate in Section 5, we take the maximum a posteriori estimate. In Section 5, we vary the model dimension $p$ by varying $F$ since $p = 2F + 2$.

## E.2  Portfolio optimization details

We assume $x$ is drawn from a multivariate Gaussian distribution with unknown covariance matrix for $D = 25$ and $N = 1000$ datapoints. We randomly sample the true covariance matrix using a Lewandowski-Kurowicka-Joe distribution on positive definite matrices. We evaluate MC-EIF and the one step estimator using the relative expected volatility (REV) and the root mean-squared-error (RMSE) between the estimated and the true optimal portfolio weights. Here, the expected volatility is calculated by applying the estimated weights with the actual covariance to the objective in Equation 9. Repeating our experiment using 50 randomly generated datasets, we find that MC-EIF enables substantially improved estimates, as shown in Table 1.

### E.3 Additional Figures

Here, we provide experimental results that provide interesting insight, but do not directly support the key claims of our paper.

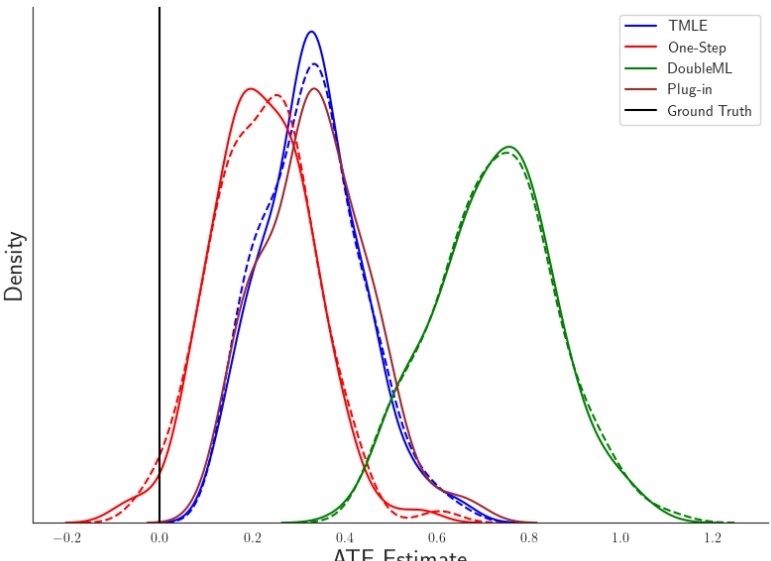

Figure 5: **Comparison of plug-in estimator and efficient estimators using MC-EIF and analytic EIF for estimating ATE with synthetic data.** The true ATE is 0. Closer to zero the better. The distribution is over 100 simulated datasets. Dashed lines represent the estimates using the analytic EIF, and the solid lines represent using MC-EIF (when applicable). Given the high-dimensionality of the problem, the estimation leads to non-zero centering (i.e., some bias remains even after influence function based corrections). Importantly, this is a property of the influence corrected estimators, and is not an artifact introduced by MC-EIF. Instead, we chose our empirical study to demonstrate that MC-EIF produces near-identical results for a diversity of statistical tasks, with a diversity of statistical implications.

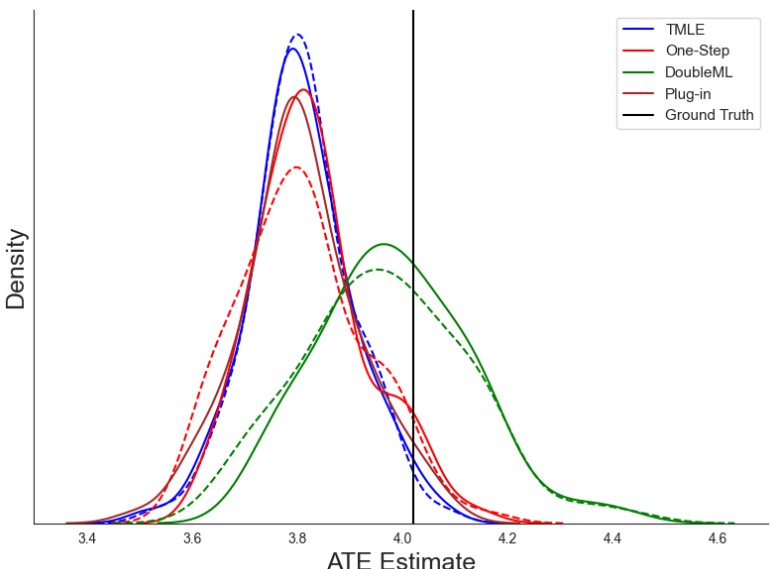

Figure 6: **Comparison of plug-in estimator and efficient estimators using MC-EIF and analytic EIF for estimating ATE with real data.** Here, we use the Infant Health and Development Program semi-synthetic data [Hil11] commonly used for effect estimation benchmarking with the same causal GLM as our synthetic data experiments. Here, MC-EIF produces estimates that are closely aligned with the analytic EIF estimators and, in the case of double machine learning, produce estimates that are much closer to the true ATE. Again, we emphasize that the choice of influence corrected estimator is separable from the choice of how to estimate the efficient influence function, which is our focus in this work.

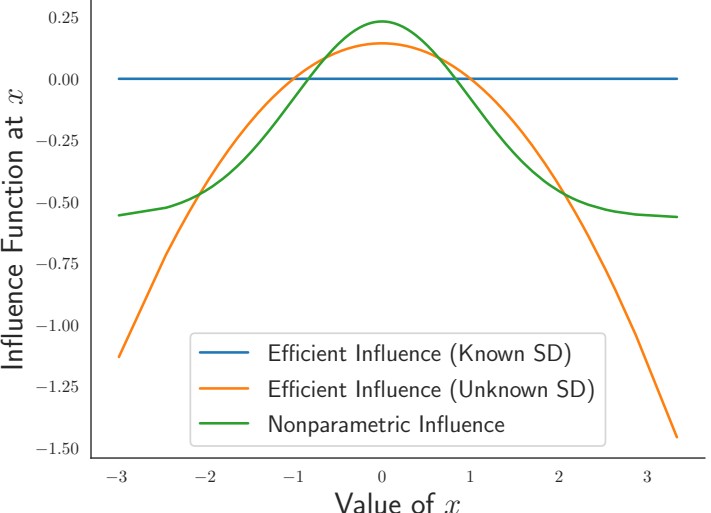

Figure 7: Nonparametric and efficient influence functions for expected density.

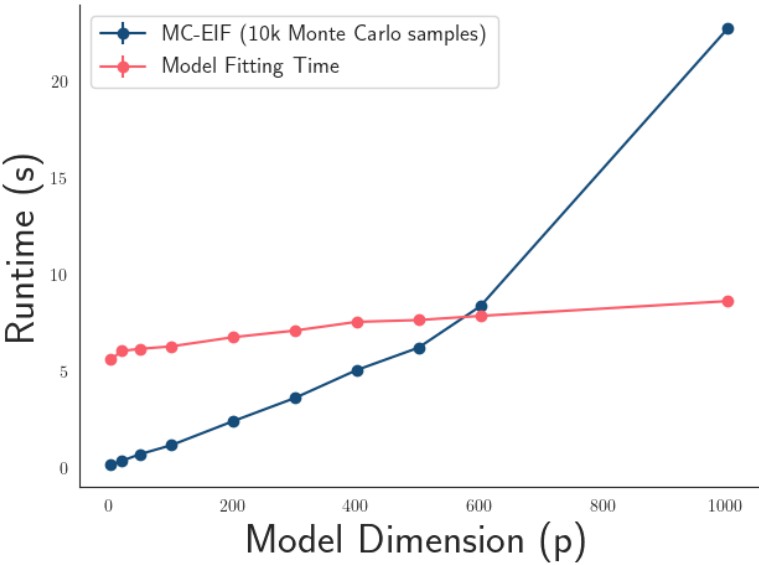

Figure 8: Runtime of fitting point estimate and computing MC-EIF as a function of model size.

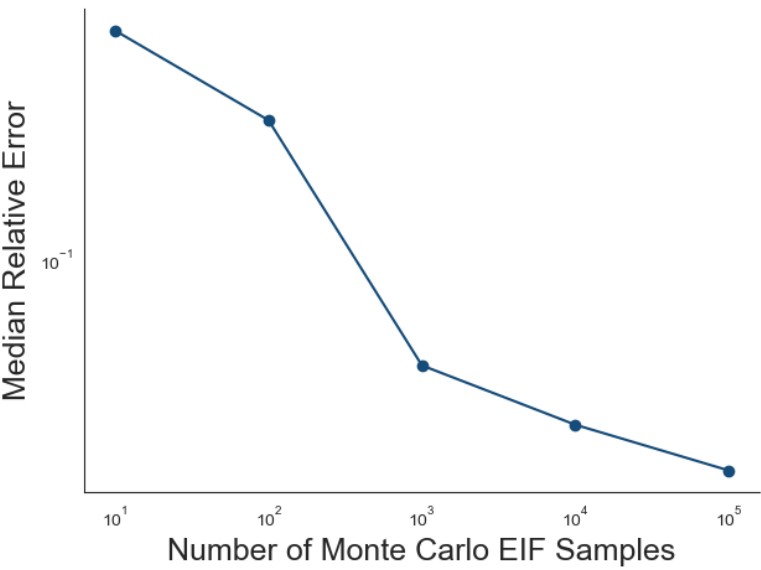

Figure 9: Median relative error between MC-EIF and true efficient influence function for unknown variance model and expected density functional. Median absolute error computed by randomly sampling points to evaluate EIF, computing the relative error at each point, and then taking the median.

