# OpenReview forum: "Automated Efficient Estimation using Monte Carlo Efficient Influence Functions"
_NeurIPS.cc/2024/Conference — NeurIPS 2024 spotlight_

### Official Review · Reviewer_hhJt · 2024-07-08

**Soundness:** 3
**Presentation:** 2
**Contribution:** 3
**Rating:** 7
**Confidence:** 1

**Summary:**

Efficient influence functions (EIFs) for nonparametric estimands are used to construct debiased estimators. Existing methods are primarily estimand-specific and require intricate analytic derivations; existing automated methods don't scale well. This paper proposes general Monte-Carlo estimators of the efficient influence function (MC-EIF) that to be used within a probabilistic programming workflow, shows their convergence to the true efficient influence function, as well as convergence of some estimates based on MC-EIF to the true value.

Disclaimer: I had little familiarity with the use of influence functions for constructing estimators before reading this paper.

**Strengths:**

The motivation for the paper is clear, and the developed mechanism is sound, and the convergence results are convincing. The extensions of the use of EIF for estimation that are opened with the proposal of the estimator are intriguing.

**Weaknesses:**

* I found the paper was harder to understand than was necessary due to unclear notation. In the Problem Statement, P and Q are used interchangably to mean the pdf or the probability measure (e.g. in Definition 2, it's a measure with respect to which the L2 space is defined, and a function in said L2 space).
* The Assumptions (esp 3.1-3.3) are not easily interpretable and are not discussed; as the paper proposes a practical method, it would be made stronger by a discussion of the assumptions.

**Questions:**

* The maximum eigenvalue term in Theorem 3.8 may grow in N. Is there an assumptions making sure it does not?

**Limitations:**

* Authors do not discuss if there is value for lifting the assumptions of the method.

---

> ### Author Rebuttal · Authors · 2024-08-07
>
> We are very grateful to receive such positive feedback that our submission is well motivated, theoretically sound, the evidence is convincing, and that MC-EIF opens up future interesting directions. We also appreciate the actionable feedback.
>
> **(1) Notation**. We thank the reviewer for flagging this issue. To improve clarity, we will carefully clean up notation, including not using P and Q interchangeably to mean the pdf or the probability measure.
>
> **(2) Clarity of Assumptions**. We thank the reviewer for this honest feedback. We will include a more extensive discussion of assumptions in the Appendix, and clarify Assumptions 3.1-3.3 in the main body. In summary, Assumption 3.1 states that the parametric model’s probability density is continuous and differentiable with respect to its parameters, Assumption 3.2 states that the pushforward of the parametric model through the target functional is continuous and differentiable with respect to the model parameters, and Assumption 3.3 states that the Fisher information is invertible. Without these assumptions, the terms in Equation (1) do not have any meaning.
>
> **(3) Maximum eigenvalue growth**. The maximum eigenvalue of $\Sigma^{-1}$ does not grow with $N$ (the number of datapoints); $\Sigma = cov(\tilde{x})$, where $\tilde{x}$ is the normalized score vector associated with $P_{\phi}$ (defined in Assumption 3.5) which is independent of $N$. This term could, however, increase with the model dimension $p$. We could add an assumption that $\Sigma^{-1}$ is uniformly bounded but we proved the theorem for the more general case when $\Sigma^{-1}$ is not. There are of course many instances when the maximum eigenvalue of $\Sigma^{-1}$ is indeed uniformly bounded in $p$ (in which case our approximation error bound gets better). For example, if  $\Sigma$ is the identity matrix, then the maximum eigenvalue is 1 for all p.
>
> **(4) Discussion of Limitations**. While the assumptions we make throughout the paper permit a very broad class of practical models and functionals, as we discussed in Section 3, there are certainly some circumstances where relaxing the assumptions and requirements would be of practical value. For example, one may wish to use MC-EIF on functionals in models that only contain an implicit likelihood, such as Bayesian variants of off-the-shelf physics emulators (See https://www.pnas.org/doi/10.1073/pnas.1912789117). Extending MC-EIF to these models, or similarly implicit functionals, is an exciting area of future work.

---

> > ### Comment · Area_Chair_c1Ak · 2024-08-12
> >
> > Dear Reviewer hhJt:
> >
> > Can you please respond to the rebuttal as soon as possible? Your comments will be greatly appreciated. Many thanks,
> >
> > AC

---

### Official Review · Reviewer_cWd8 · 2024-07-09

**Soundness:** 3
**Presentation:** 4
**Contribution:** 4
**Rating:** 8
**Confidence:** 4

**Summary:**

The paper establishes a novel method for estimating the efficient influence functions under mild assumptions, called Monte Carlo Efficient Influence Functions(MC-EIF). The method is easy to apply and flexible in many cases, where it can seamlessly equip an existing/popular efficient estimator.

**Strengths:**

It is a really well-written paper with complete model establishment, theoretical results and fair comparisons between their MC-EIF method and other existing methods. Details are listed, like the different ways to generate the estimator and how it is sensitive to their methods and also the limitations on the dimension of model size are discussed. This MC-EIF is easy and flexible to use in many scenarios, so it gets good prospects in the application area.

**Weaknesses:**

When applying new methods in practice, especially in high-dimensional cases, the time cost is also a necessary aspect that needs to be considered, so it would be good to show the time cost for your methods and how it compares to other existing methods.

Some settings in the experiments are not listed clearly, like what is the value of $D$ and $p$ for experiments in Figure 1,2,3 or provided the value of $F$ instead?

**Questions:**

It draws my attention that, from Assumption 3.7 and Theorem 3.8, as the model size $p$ becomes larger, the constant of the bound of error will increase as $\sqrt{p \log p}$. So, when dealing with the high-dimensional problem, it may require people to use a really large number of samples ($M$) to get a desirable accuracy and the cost will be tremendous, which could be a potential drawback of this method, what do you think?

**Limitations:**

I am not sure what is the $p$ tested here, but from Figure 2, it seems like $p_{\max}$ is only $1000$, which is not a very large model. People should test on much higher dimensional problems to defend their methods. Moreover, it would be worth looking at more challenging problems, other than well-defined Gaussian problems.

---

> ### Author Rebuttal · Authors · 2024-08-07
>
> We are very grateful to receive such positive feedback on our submission’s exposition, the quality of our theoretical and empirical evidence, and MC-EIF’s generality and ease-of-use. Thank you also for helping us to improve our work with actionable feedback.
>
> **(1) Time comparison relative to other methods**. Before MC-EIF, the only general-purpose method for approximating efficient influence functions was provided by JWZ22a, who use a finite-differencing approximation. This runtime of this method is exponential with respect to dimension p with a large constant. The primary benefit of MC-EIF over these existing techniques is its substantially faster runtime. In fact, in our experiments, we could not even compare against the method in JWZ22a because the runtime was too slow for p larger than 2. That is why we only compare against JWZ22a in experiment 1 and not the other experiments where p is in the hundreds or a thousand. In Figure 7 in the Appendix we provide the runtime of our method as a function of p. We will emphasize how MC-EIF is faster than existing approaches much more prominently in the revision.
>
> **(2) Missing labels**. We will add these missing labels in the revised submission and make the choices clearer in the main body. In Figure 1(a) D=1 and p=1 (only unknown mean parameter). In Figure 1(b), D=1 and p=2 (unknown mean and variance parameter). In Figure 2, F=200, D=202 and p=402. In Figure 3, D varies from 1 to 502 and p is shown on the y-axis.
>
> **(3) Approximation quality for large p**. The reviewer is correct that the error bound grows like $\sqrt{p \log p}$. To ensure an accurate error bound, that means the number of Monte Carlo samples $M$ must scale linearly with $p \log p$. In practice, the runtime is not prohibitively slow. For example, in Figure 7 in the Appendix, we compare the runtime of fitting the model relative to computing MC-EIF as a function of model size. We see that when $p$ is less than 600, MC-EIF is faster than the time to fit the model. When $p$ is 1,000 then our implementation of MC-EIF takes 20 seconds to run on a standard M2 laptop.
>
> The computational bottleneck of MC-EIF is computing the Fisher information matrix. Fortunately, given the importance of the Fisher information matrix in statistics, there are many ways to speed up computations here. See, for example, “A Kronecker-factored approximate Fisher matrix for convolution layers”. We will include a discussion of how these approximations can be used with MC-EIF in the paper.
>
> **(4) Larger p problems / real data**. Relative to existing work, we evaluated our method on much larger p problems. For example, in JWZ22a, the authors only evaluated on settings with p=1, as their method’s runtime scales exponentially in p. Nevertheless, we can certainly evaluate our method on a larger p problem in the revised submission. Since submission, we have evaluated MC-EIF on real, non-Gaussian data and found similar empirical results on synthetic data in Section 5 of our paper. We will include these results in the revised submission.

---

> > ### Comment · Reviewer_cWd8 · 2024-08-09
> > **Response to rebuttal**
> >
> > Thank you very much for your responses! My main concern revolves around the time scale of the methods, which I believe is a key strength. However, this aspect hasn't been sufficiently highlighted in the paper. I hope the author can give this more attention in the revision and also follow through on addressing the larger problem as promised. Overall, I will maintain my score.

---

### Official Review · Reviewer_ebo4 · 2024-07-12

**Soundness:** 3
**Presentation:** 3
**Contribution:** 3
**Rating:** 6
**Confidence:** 2

**Summary:**

The paper proposes Monte Carlo Efficient Influence Functions (MC-EIF), an automated technique for numerically computing efficient influence functions using existing differentiable probabilistic programming systems. MC-EIF simplifies efficient statistical estimation for high-dimensional models, achieving optimal convergence rates and consistency without the need for complex manual derivations. The approach is validated both theoretically and empirically.

**Strengths:**

The paper is well-written and can be followed easily. The authors introduce Monte Carlo Efficient Influence Functions (MC-EIF), a technique for numerically computing EIFs using existing AD and PPL system quantities. They express EIFs as a product of the gradient of the functional, the inverse Fisher information matrix, and the gradient of the log-likelihood. This method automates the construction of efficient estimators, avoiding manual derivations, and provides accurate, generalizable estimates applicable to various functionals and models. They also provide a non-asymptotic error bound on the quality of their approximation, showing how estimators using MC-EIF achieve the same asymptotic guarantees as using analytic EIFs. Empirical results show MC-EIF outperforms existing approaches without degrading estimation accuracy.

**Weaknesses:**

I am not fully familiar with this line of work, so I am unable to identify a major weakness. However, I have some questions regarding the assumptions  and the effectiveness of the proposed approach on real datasets, which I have added to the Question sections.

**Questions:**

Q1. In Assumption 3.5, authors assume that the normalized score vector is sub-Gaussian with a parameter $C_1$. I want to know if this constant scales with respect to the dimension $D$ and model size $p$. If yes, what is the scaling? If no, can you clarify why it does not scale?

Q2. The authors assume that the map $\phi$ to $P_{\phi}$ is continuous. My question is, in order to approximate the Fisher information $\hat{I_{m}}$, do we need to know what $P_{\phi}$ actually is?


Q3. The authors back up their theoretical results with synthetic data experiments. Given the assumptions, it is not clear how applicable the proposed method is to real datasets, and how should the results be interpreted if these assumptions don't hold?"

**Limitations:**

The authors clearly mention the assumptions before stating their theoretical guarantees. Also, the assumptions are explained clearly. I don't see any potential negative societal impact of their work.

---

> ### Author Rebuttal · Authors · 2024-08-07
>
> We are very grateful to receive such positive feedback on our submission’s exposition, and empirical and theoretical evidence. We are also very grateful to receive actionable feedback we can use to further improve our work.
>
> **(1) Assumption 3.5**. The constant C will not grow with D in Assumption 3.5. If there exists a universal constant k such p <= kD, then C will not grow with p either but this is not a necessary assumption. We will include this discussion in the revision but for completeness we provide a quick sketch below:
>
> The score vector $\nabla \log P_{\phi}(x)$ is a p-dimensional vector so its expected squared norm grows linearly with $p$. The expected squared norm of the normalized score $\frac{1}{\sqrt{D}} \nabla \log P_{\phi}(x)$ scales as $O((\frac{1}{\sqrt{D}})^2 \times p) = O(p/D)$. Hence, if $p \leq kD$, then the expected squared norm is bounded. Hence, the sub-Gaussian constant in Assumption 3.5 for the normalized score will not grow with p or D.
>
> **(2) Continuity assumption**. MC-EIF addresses the problem of efficient estimation with parametric models using efficient influence functions. Here, $P_{\phi}$ is that parametric model itself, and is known by construction for any fixed set of parameters $\phi$. For example, if we wish to use MC-EIF to construct an efficient estimator for a linear regression model with Gaussian errors, then $\phi$ would be the fitted regression coefficients and $P_{\phi}$ is the induced Gaussian distribution over outcomes. We will add a note clarifying this in the revision.
>
> **(3) Real data**. Since submission, we received other feedback to add real data experiments so we evaluated MC-EIF on real data from the UCI machine learning repository. We find that the approximation quality is similar to the good performance on synthetic data in Section 5 of our paper! This is not surprising, as previous work on efficient estimation has already demonstrated the benefit of influence function based estimation with real data, and MC-EIF closely approximates the efficient influence function.

---

> > ### Comment · Reviewer_ebo4 · 2024-08-08
> > **Re.**
> >
> > Thanks for the response to my questions. A clarifying note on the Continuity assumption, as the authors mentioned, would be very useful. I still believe that the paper would benefit from some real data experiments or results. Overall, I will keep my score for the paper.

---

### Decision · Program_Chairs · 2024-09-25

**Decision:**

Accept (spotlight)

**Comment:**

This paper develops a general Monte-Carlo approach for approximating efficient influence functions. It aims to provide an automated procedure within an existing probabilistic programming workflow. It also provides non-asymptotic error bounds on the quality of the Monte-Carlo approximation and examines three specific estimators: von Mises one step estimator, debiased/double ML, targeted minimum loss estimation. The paper is interesting and fits well with the current developments in the literature. It would be useful to revise the paper for the following aspects: (i) real-data experiments, which the authors indicate that they already carried out; (ii) an emphasis on the advantage in terms of computational speed; (iii) a more serious discussion of limitations (e.g., computation of the Fisher information matrix) as well as the underlying assumptions and requirements.